# *Pyricularia*’s Capability of Infecting Different Grasses in Two Regions of Mexico

**DOI:** 10.3390/jof9111055

**Published:** 2023-10-27

**Authors:** Ivan Sequera-Grappin, Elsa Ventura-Zapata, Erika Alicia De la Cruz-Arguijo, Claudia Patricia Larralde-Corona, Jose Alberto Narváez-Zapata

**Affiliations:** 1Instituto Politécnico Nacional, Centro de Biotecnología Genómica, Blvd. del Maestro S/N Esq. Elías Piña. Col. Narciso Mendoza, Reynosa C.P. 88700, Tamaulipas, Mexico; isequerag1400@alumno.ipn.mx (I.S.-G.); edelacruz@ipn.mx (E.A.D.l.C.-A.); plarralde@ipn.mx (C.P.L.-C.); 2Instituto Politécnico Nacional, Centro de Desarrollo de Productos Bióticos, Ctra. Yautepec-Jojutla, Km.6, calle CEPROBI No. 8, Col. San Isidro, Yautepec C.P. 62731, Morelos, Mexico; eventura@ipn.mx

**Keywords:** gray leaf spot, pyricularia, PCR fingerprinting, grasses

## Abstract

The genus *Pyricularia* includes species that are phytopathogenic fungi, which infect different species of Poaceae, such as rice and sorghum. However, few isolates have been genetically characterized in North America. The current study addresses this lack of information by characterizing an additional 57 strains of three grasses (*Stenotaphrum secundatum*, *Cenchrus ciliaris* and *Digitaria ciliaris*) from two distant regions of Mexico. A *Pyricularia* dataset with ITS sequences retrieved from GenBank and the studied sequences were used to build a haplotype network that allowed us to identify a few redundant haplotypes highly related to *P. oryzae* species. An analysis considering only the Mexican sequences allowed us to identify non-redundant haplotypes in the isolates of *C. ciliaris* and *D. ciliaris*, with a high identity with *P. pennisetigena*. The Pot2-TIR genomic fingerprinting technique resulted in high variability and allowed for the isolates to be grouped according to their host grass, whilst the ERIC-PCR technique was able to separate the isolates according to their host grass and their region of collection. Representative isolates from different host grasses were chosen to explore the pathogenic potential of these isolates. The selected isolates showed a differential pathogenic profile. Cross-infection with representative isolates from *S. secundatum* and *C. ciliaris* showed that these were unable to infect *D. ciliaris* grass and that the DY1 isolate from *D. ciliaris* was only able to infect its host grass. The results support the identification of pathogenic strains of *Pyricularia* isolates and their cross-infection potential in different grasses surrounding important crops in Mexico.

## 1. Introduction

*Pyricularia* species, particularly *P. oryzae* (the teleomorph of *Magnaporthe oryzae*), cause gray leaf spot (GLS) disease in more than fifty species of Poaceae, including in economically and agriculturally important crops, such as sorghum (*Sorghum bicolor*) and rice (*Oryza sativa*), and in grasses surrounding crop fields [1]. In Mexico, there have been reports of GLS on rice and on buffel grass (*Cenchrus ciliaris*) and *Digitaria ciliaris* [2,3,4]. More broadly, in the Americas, diverse *Pyricularia* isolates have been reported to infect *Stenotaphrum secundatum* grass [5], *Cenchrus grass* [6], *Festuca arundinacea* Shreb. [7] and *Hakonechloa macra* (Japanese forest grass) [8]. *P. oryzae* has a broad host range (Poaceae), which may allow pathogens to shift to rice and neighboring plants [1]. The cross-infectivity of isolates from rice and other Poaceae species implies that host shifts in *P. oryzae* and related species may occur in rice and other plants surrounding rice fields [9,10]. Thus, host divergence can increase the genomic complexity of isolates [10]. In addition, the expansion of *P. oryzae* and its host shifting have been widely reported and may be implicated in GLS outbreaks in important crops [1]. Therefore, *Pyricularia* strain characterization in grasses might also be important to prevent future outbreaks of GLS disease in crops.

*Pyricularia* species are difficult to control due to the complex diversity and wide geographic distribution of their isolates [11], and this problem has also been observed in isolates with different phenotypic profiles [12], locations of collection [13] and crop varieties [14]. Different studies have related genetic variability to the pathogenic profile for *Pyricularia* isolates [9,15,16]. Genetic variation may be caused by a high copy number of transposons, such as Pot2-TIR, which is often used in *P. oryzae* strain analyses [17], or by randomly dispersed and non-repetitive ERIC regions [18], which have also been useful in assessing genetic diversity in fungal species [18,19].

In Mexico, there are no reports on the genetic variability or the potential of cross-infection of *Pyricularia* strains in grasses surrounding crops. Therefore, the objective of this study was to isolate and characterize *Pyricularia* isolates infecting different grasses (*S. secundatum*, *C. ciliaris* and *D. ciliaris*) in localities surrounding sorghum and rice fields in Mexico.

## 2. Materials and Methods

### 2.1. Fungal Isolation and General Characterization

Isolates were obtained from infected leaf tissues of *S. secundatum*, *C. ciliaris* and *D. ciliaris*. Host identification was undertaken according to the hosts’ morphologic characteristics, using the web catalogue of Mexican weeds provided by CONABIO (http://www.conabio.gob.mx/malezasdemexico/2inicio/home-malezas-mexico.htm, accessed on 10 October 2023). Sampling was conducted in four localities in two Mexican states (Morelos and Tamaulipas), in which the pathogens considered in this study cause persistent infections in the hosts mentioned above. These Mexican states have different environmental conditions and are geographically distant from each other (~1000 km). The sampling sites were Yautepec (Morelos; 18.51 N-99.04 W), with an altitude of 1210 m; Zacatepec (Morelos; 18.39 N-99.11 W), with an altitude of 917 m; Oacalco (Morelos; 18.92 N-99.03 W), with an altitude of 1243 m; and Reynosa (Tamaulipas; 25.58 N-98.16 W), with an altitude of 33 m. These locations were selected because they are mainly agricultural, with rice being the main crop in the Morelos locations and sorghum being the main crop in the Tamaulipas location.

Plant samples associated with the main crops in the regions mentioned above and that showed disease symptoms were collected at a separation distance of at least 100 m. Immediately after collection, small pieces (≈0.5 cm × 0.5 cm) were cut from the infected tissue close to the damaged areas. The pieces were submerged for 10 min in a solution of 0.1% NaClO for surface sterilization, rinsed three times with distilled water and attached via surface tension to the lids of Petri dishes (50 mm) containing a water agar medium (2% *w*/*v*). The plates were incubated at 28 °C ± 1 °C in darkness for 24–48 h. Mycelia growing on the cut pieces were observed through an optical microscope for conidia identification. The plates with abundant conidia formation were gently stirred to release the spores and place them over the medium. The lids were changed for new sterile ones and the plates were incubated under the same conditions mentioned above for 3 days.

A microscopic inspection was conducted to identify non-grouped conidia on the medium surface and the surrounding area was labeled on the Petri lids. Agar plugs containing this area were transferred to a potato dextrose agar medium (PDA, Difco, Becton Dickinson and Co., Holdrege, NE, USA) and incubated for 7 days. The cultures were then sub-cultivated in a V8 agar medium and incubated for 2 weeks under the same conditions mentioned above. Plugs of 5 mm diameter containing mycelia and conidia were placed in 2 mL cryogenic vials containing 1 mL of glycerol solution 10% (*v*/*v*). The vials were stored at −20 °C. Morphological structures (conidia morphology, color and septa number) were documented using fluorescent microscopy with 40× and 100× zooms (Olympus BX-51; Olympus, Life Science Research, Baltimore, MD, USA). Image-Pro Express software package ver. 6.3 (Media Cybernetics, Silver Spring, MD, USA) was used for phenotypic (qualitative) assessment and average size (quantitative) evaluation. Isolates were labeled according to their host and location.

Mating type characterization was conducted by using the primer sets MAT1-1 and MAT1-2 reported in [20]. PCR was performed in a total volume of 25 µL. The reaction consisted of 50 ng of genomic DNA, obtained as described below; 200 mM of each primer; 0.8 mM dNTPs (Bioline, Memphis, TN, USA); 1X PCR buffer (Bioline, Memphis, TN, USA); 1.5 mM MgCl_2_ (Bioline, Memphis, TN, USA); and 1U of Taq DNA polymerase (Bioline, Memphis, TN, USA). The PCR conditions consisted of initial denaturation for 5 min at 95 °C, 35 cycles of 30 s of denaturation at 95 °C, 30 s of annealing primers at 54 °C, 1 min of extension at 72 °C and, finally, an extension at 72 °C for 7 min. The PCR products were separated via electrophoresis in an agarose gel (1% *w*/*v*), and they were visualized using SYBR Green (Invitrogen, Waltham, MA, USA).

The fertility status of these isolates was also assayed by pairing them with opposite mating-type strains on oatmeal agar plates, with two mycelium plugs (0.5 cm) placed on opposite sides for 4 weeks under environmental conditions. Compatibility was assessed by searching for the presence of reproductive structures (perithecia) under a microscope, as described above.

### 2.2. ITS Amplification and Sequencing Analysis

Isolated strains were cultured in Petri dishes (9 cm diameter) containing PDA (Difco, Becton Dickinson and Co., Holdrege, NE, USA) at 28 °C ± 1 °C for 7 d. DNA was extracted in 1.5 µL tubes according to the CTAB protocol described in [21]. Each DNA sample was quantified using a spectrophotometer (NanoDrop 2000; Fisher Scientific Inc., Carthage, MO, USA). ITS amplification was conducted on a thermocycler (2720 Thermal Cycler; Applied Biosystems, Waltham, MA, USA) with the universal primers ITS1 (5′ TCCGTAGGTGAACCTGCGG 3′) and ITS4 (5′ TCCTCCGCTTATTGATATGC 3′) [22]. PCR was performed in a total volume of 25 µL. The reaction consisted of 50 ng of genomic DNA, 200 mM of each primer, 0.8 mM dNTPs (Bioline, Memphis, TN, USA), 1X PCR buffer (Bioline, Memphis, TN, USA), 1.5 mM MgCl_2_ (Bioline, Memphis, TN, USA) and 1U of Taq DNA polymerase (Bioline, Memphis, TN, USA). The PCR conditions consisted of initial denaturation for 5 min at 94 °C, 35 cycles of 30 s of denaturation at 94 °C, 30 s of annealing primers, 1 min of extension at 72 °C and, finally, an extension at 72 °C for 7 min.

The PCR products were separated via electrophoresis in 1% (*w*/*v*) agarose and visualized using SYBR Gold (Invitrogen, Waltham, MA, USA). The PCR products were purified with a kit according to the manufacturer’s protocol (BioRad, Hercules, CA, USA). After purification, the PCR products were directly sequenced using both original primers. Sequencing was conducted by Macrogen Inc. (Seoul, Republic of Korea). A Blastn analysis was conducted on all sequences obtained. Nucleotide sequences were deposited in the GenBank database, as summarized in Table 1.

### 2.3. Haplotype Network Analysis

All the ITS sequences from the *P. oryzae* strains were retrieved from GenBank by using the “refdb” packages in Rstudio ver. 4.3.0. Sequences without longitude/latitude data were removed from the dataset to assign localization to all accessions. The database was updated with the sequences of this study and manually revised to avoid entry mistakes (Appendix A). Hotspots of the isolations were visualized in Rstudio by using the “ggplot2 ver. 3.4.2”, “sf ver. 1.0-14” and “spData ver. 2.3.0” packages. Then, the “Biostrings ver. 2.68.1” and “ape ver. 5.7-1” packages were used to obtain and write the fasta sequences of the final database. These fasta sequences were used to build a Clustalw alignment by using the “msa ver. 1.32.0” package. Trimming was conducted on sequence alignments to avoid sequence termination mistakes in all sequences analyzed. The trimmed ITS region had a length of 416 bp, which was localized, according to the reference *P. oryzae* (MT757299; MoK19-32) strain, between 87 bp and 503 bp. Then, the “haplotypes ver. 1.1.3.1” and “pegas ver. 1.2” packages were used to build a haplotype network analysis in Rstudio ver. 4.3.0. To simplify the haplotype links on the map, a threshold of 55–60 was selected.

### 2.4. Genomic Fingerprinting

Two methods were used for genome-wide characterization. First, Pot2-TIR amplifications were performed in a final volume of 25 µL. The Pot2.TIR profile was generated according to [23], using a single external primer: Pot2-TIR (5′ ACAGGGGGTACGCAACGTTA 3′). Second, ERIC-PCR amplification was conducted according to the method developed in [18], with the primers ERIC1R (5′ ATGTAAGCTCCTGGGGATTCAC 3′) and ERIC2I (5′ AAGTAAGTGACTGGGGTGAGCG 3′). For each reaction, the following were added: 50 ng of genomic DNA, 0.5 µM of each primer, 200 µM of dNTPs, 1× of reaction buffer, 1.5 mM of MgCl_2_ and 2.5 U of Taq polymerase. PCR conditions were initial denaturation for 5 min at 94 °C, 35 cycles of 30 s of denaturation at 94 °C, 1 min of annealing at 58 °C (Pot2-TIR)/49 °C (ERIC-PCR), 4 min of extension at 65 °C and a final extension for 7 min at 72 °C. PCR fingerprints were analyzed in 2% agarose gels, stained with SYBR Gold (Invitrogen, Waltham, MA, USA) and run in 0.5% TBE buffer (89 Mm Tris pH 7.8, 89 Mm boric acid, 2 Mm EDTA) at 80 V for 6 h and 4 h for Pot2-TIR and ERIC-PCR, respectively. Polymorphic profiles were visualized with UV light. Genetic similarity among isolates was determined using a score given for the presence (1) or absence (0) of bands of a particular molecular weight. A binary matrix was made for each fingerprinting marker. A cluster analysis conducted using complete-linkage clustering was performed using the UPGMA algorithm in Rstudio ver. 4.3.0 [24]. Experiments were conducted in triplicate.

### 2.5. Infection Assay Using Spot Inoculation

Fungal isolates were grown on a water agar medium (2% agar; *w*/*v*). The isolates were incubated at 28 °C ± 1 °C for approximately one month before inoculation. A pathogenicity assay was conducted on the leaves of *S. secundatum*. This grass was selected to set up the pathogenic assay since it occurs in all study locations. *S. secundatum* plants collected in Yautepec (Morelos) were grown at temperatures of 28 °C to 30 °C for 30 days in pots containing peat moss, compost and perlite at a ratio of 40:40:20. The plants were periodically irrigated with half-strength Hoagland solution. The plants were propagated by short, branched rhizomes.

Pathogenicity assays were conducted as follows: Leaf segments of 5 cm were disinfected with 0.1% NaClO for 10 min, rinsed 3 times with distilled water and dried with sterile paper. Each leaf segment was placed on a glass microscope slide (2.5 cm × 8 cm) and the ends of the leaf segment were attached to the slide with tape. The microscopy slides with the leaf tissues were placed in Petri dishes of 9 cm containing water agar (2% *w*/*v*) supplemented with chloramphenicol (400 mg/L) and cycloheximide (400 mg/L). Undamaged leaf segments were inoculated by placing 5 mm diameter plugs in their centers, obtained from the water agar inoculum plates containing mycelium and conidia (20–30 conidiophores/plug).

To compare the infection capabilities of the fungal isolates on already damaged foliar tissue, the same method as mentioned above was used, but by damaging the leaf surface via punction [16]. Using a sterile pipette, a spot (≈1 mm of diameter) was crushed in the center of *S. secundatum* grass segments, pressing just enough to injure the tissue without punching out a hole. Finally, medium plugs containing mycelia were placed on the punched areas, as mentioned above. Plates were placed at 25 °C ± 1 °C for 7 d under short-day conditions (8 h light/16 h dark). The percentage of foliar damage area was analyzed using Image J ver. 1.53e software with the default RGB threshold color settings (≈45, 45, 0). A rating scale was developed for an evaluation of the damage based on previous studies performed on rice detached leaves [25] and on Italian ryegrass (*Lolium multiflorum* Lam.) [26]. These methods use a damage score from zero to four according to the specific type of lesion. Four lesion scales, according to the extent of the damaged area, were applied. A detailed description is given in the Results Section. For each treatment, five replicates were performed.

### 2.6. Cross-Infectivity Assay

Representative fungal isolates were inoculated on three grass species (*C. ciliaris*, *D. ciliaris* and *S. secundatum*). These are wild grasses susceptible to *Pyricularia* infection that grow in the same regions where sorghum and rice are cultivated in Mexico. *S. secundatum* grass was obtained from all study locations. *C. ciliaris* grass was obtained from the locations of Yautepec (Morelos) and Reynosa (Tamaulipas). *D. ciliaris* was only obtained from the Yautepec (Morelos) location. The grasses were morphologically identified by using the CONABIO web catalogue (http://www.conabio.gob.mx/malezasdemexico/2inicio/home-malezas-mexico.htm, accessed on 10 October 2023). The grasses were grown at temperatures of 28 °C to 30 °C for 30 days in pots containing peat moss, compost and perlite at a ratio of 40:40:20. The plants were periodically irrigated with half-strength Hoagland solution. Experiments were performed in triplicate, and a water control was included. Sporulation was induced in a V8 agar medium via incubation for 20 days with light and dark cycles of 12 h at 28 °C. A spore suspension (1 × 10^5^ spores/mL) was prepared with 0.02% and 0.25% Tween 20 (Fisher Scientific, Carthage, MO, USA) and gelatin (Duche, CDMX, Mexico), respectively [25]. The leaves were inoculated with 30 mL of a conidial suspension using an airbrush. The plants were immediately covered with dark plastic bags at 28 °C for 48 h. Then, the plants were exposed to light and dark cycles of 12 h at 28 °C for 7 days. Disease occurrence was photographically recorded in 20 complete leaves with evident disease symptoms. The percentages of foliar damage area were analyzed using Image J software, as mentioned above. The scale in cm was set up by using the function “set scale” in Image J ver. 1.53e. Damage evaluation was rated on the Standardized Evaluation System for rice (SES scale) from 0 to 9, as described in [27].

### 2.7. Statistical Analysis

The data were analyzed using an analysis of variance (ANOVA) with a significance value of α = 0.05. A one-way ANOVA was performed on the damaged area using plug inoculation. A two-way ANOVA was performed if the punction treatment had an effect on the virulence level and infection profiles. A Tukey test (α = 0.05) was performed to determine the differences between the infection patterns. All statistical analyses were performed by using Minitab 17 software.

## 3. Results

### 3.1. Isolation and General Characterization

Fifty-seven *Pyricularia* isolates were collected from *S. secundatum*, *C. ciliaris* and *D. ciliaris* grasses in four localities in the states of Morelos and Tamaulipas. In general, the conidia obtained on PDA and V8 agar media showed a pyriform morphology, narrowed toward the tip and rounded at the base; colorless; smooth with three septa; and with an average size of 28.1 µm and 8.5 µm in length and width, respectively (Appendix A). The mating type analysis showed PCR products in the *P. oryzae* isolates of 809 bp and 940 bp for *MAT1-1* and *MAT1-2* alleles, respectively. The *MAT1-1* allele was found in almost all *P. oryzae* isolates analyzed. Only seven isolates (SR1, SRf 1, SRf4, SRf6 SRf10, SRf11 and SRf12) collected from *S. secundatum* in Reynosa were *MAT1-2* positive (Table 1 and Appendix A). The fertility status of these isolates according to their *MAT-1* allele was examined, but no reproductive structures were found under the culture conditions evaluated. The *Pyricularia* isolates from *C. ciliaris* and *D. ciliaris* did not amplify when using this primer set.

### 3.2. ITS Sequence Analysis

A single band of around 550 bp was obtained via PCR amplification using the ITS1/ITS4 primers. These experimental sequences were added to the ITS database built with the sequences retrieved from GenBank, resulting in 427 sequences (Appendix A). The initially retrieved sequences resulted in 34,709 accessions. However, most of them had neither longitude nor latitude data in their metadata; therefore, they were discarded. The hotspot analysis allowed us to determine that the sequences of this study are the only ones submitted with localization data in North America (Figure 1A). 

The high variability in the lengths of the sequences retrieved from GenBank led us to select only one trimmed region (416 bp) for the haplotype network analysis. This analysis classified some redundant haplotypes (with a haplotype diversity of 0.765) for the majority of the sequences analyzed, including most of the experimental sequences obtained in the current study, and some non-redundant haplotypes in accessions mainly from Africa (Figure 1B). Redundant haplotype 1 was found in almost all sequences collected in Mexico. However, a more detailed analysis allowed us to detect other specific haplotypes (with a haplotype diversity of 0.716) for the grasses *C. ciliaris* and *D. ciliaris* (Figure 1C). Specifically, *D. ciliaris* haplotypes were more varied, with diverse polymorphisms regarding the redundant haplotypes of the accessions isolated from *S. secundatum*. In general, the ITS sequences of the isolates of this study produced Blast identities (≥99%) with *P. oryzae* sequences, although a few sequences, isolated mainly from the *C. ciliaris* and *D. ciliaris* grasses, also exhibited a high identity with *P. pennisetigena* accessions (Table 1), and this corresponded to the non-redundant haplotypes described above (Figure 1C).

### 3.3. Genomic Fingerprinting

To gain more information on the genomic variability in the *Pyricularia* isolates collected in Mexico, two methods for genomic fingerprinting were conducted to expand the number of genomic regions of analysis. Firstly, the Pot2-TIR profiles of the *Pyricularia* isolates were analyzed (Figure 2A). A total of 19 polymorphic PCR products were revealed by using the Pot2-TIR primer, which ranged between 0.3 Kb and 3.5 Kb (Appendix A). UPGMA based on Pot2-TIR profiles shows three main clades with a high height (~3), grouping the isolates according to their host grass (marked with different colors).

This analysis also grouped the isolates according to their probable identity, separating with a high height value (~2.8) the isolates identified as *P. oryzae* (host, *S. secundatum*), *P. pennisetigena* (host, *C. ciliaris*) and *Pyricularia* sp. (host, *D. ciliaris*). Interestingly, all isolates showed a clear grouping according to their host grass, supporting their differential species identities. For example, the isolates from *C. ciliaris* that showed a high (>99%) identity with *P. pennisetigena* were clearly grouped together, supporting their probable identity as *P. pennisetigena*, and the biases of the phylogenetic analysis can clarify the identity of this cryptic species [28]. The ITS locus in combination with the 19 polymorphic Pot2-TIR bands thus might help to support interspecies identification. The Pot2-TIR profile also showed a high variability within the isolates clearly identified (>99%) with a specific species (i.e., *P. oryzae*). In the current study, the *P. oryzae* isolates from *S. secundatum* showed a high variability, with seven subclades with a high (~1) height difference. These genomic differences could also be useful in distinguishing the regions of collection or associated crop agronomical conditions. These isolates spanned five subclades: two clades with isolates only from Tamaulipas (sorghum fields), two clades with isolates only from Morelos (rice fields) and one clade with *P. oryzae* isolates from both Mexican states. Therefore, although this analysis recognized the distance between the studied regions (~1000 km), it also showed a subclade of isolates exhibiting a probable genetic environmental adaptation to the associated grass (*S. secundatum*).

Genomic fingerprinting was complemented with ERIC-PCR profiles (Figure 2B and Appendix A) as a second method. In general, this analysis showed less variability, having only 11 polymorphic PCR products. However, these were enough to more clearly separate (height ~2.2) the isolates according to the host grass, thus supporting the results previously obtained using the Pot2-TIR fingerprinting method. Interestingly, the subclades of *P. oryzae* (host *S. secundatum*) were grouped according to their *MAT1* allele, and all the isolates related to *Pyricularia* spp. and *P. pennisetigena* were unable to be amplified with the *MAT1* allele primers (Table 1 and Appendix A), which suggests a probable relation between the PCR products amplified by these randomly amplified sequences and the *MAT1* alleles. More efforts are necessary to clarify the relation between this non-repetitive region and the *MAT1* allele.

### 3.4. Pathogenicity Assay

An assay was performed on the leaves of *S. secundatum* to classify the pathogenicity of the isolates. Contrasting *P. oryzae* isolates from *S. secundatum* were selected to conduct this assay (Table 2; Figure 3). The infection method was implemented with two different treatments: agar plug inoculation with and without the infliction of puncture damage to detached leaves. In addition, a scale of the infection response was set up, modified from [25,26], as follows: Scale 0 comprises leaves with lesions without evident symptoms. Scale 1 comprises leaves with a damaged area with a diameter of less than 2 mm, and non-sporulating and dark brown lesions. Scale 2 comprises leaves with expanding dark brown and non-sporulating lesions. Scale 3 comprises leaves with a damaged area in a diamond or small circular shape with sporulating centers. Scale 4 comprises leaves with a large damaged area and expanding irregular lesions, as well as with sporulating areas (Figure 3A). 

The *P. oryzae* isolates produced different lesion types and extensions of damaged area according to the infection method used. The infection assay via plug inoculation showed characteristic GLS lesions, such as the extension of coalescent spots and sporulated lesions (Figure 3B). Agar plug inoculation without punch damage to the detached leaves showed a reaction that ranged across the whole scale, with SY2 being the most aggressive isolate (Table 2). However, when the inoculation was conducted with punction damage, all isolates showed a reaction classified as type 4. The damaged area also exhibited differential values according to the isolate and to the infection method assayed (Figure 3B). When the inoculation was conducted without punch damage, the *Pyricularia* isolates ranged between 1.6 mm^2^ and 76.7 mm^2^, with an average damaged area of 22 mm^2^.

The damaged area increased when the leaf was punctured, being up to 14.5 times larger than after the treatment without damage in the SY4 isolate, with a value of 73.7 mm^2^. In general, the damaged area had an average size of 49.6 mm^2^, an increase of 6.7 times. The Tukey test grouped these damaged areas into three groups, which separated these *P. oryzae* isolates according to their sampling location. In this last analysis, the SR1 isolate was classified into a different Tukey group (group c) with a damaged area of 42.1 mm^2^ (Table 2 and Appendix A).

Representative *Pyricularia* isolates were selected to explore their pathogenic profiles by assessing their genetic variability. Genetic and genomic relations, localities with different associated crops, and the host and *MAT1* allele were considered to choose these representative isolates. Thus, SY2 (*P. oryzae*; host, *S. secundatum*; Morelos state; and *MAT1-1* allele), SR1 (*P. oryzae*; host, *S. secundatum*; Tamaulipas state; and *MAT1-2* allele), CY1 (identified as probable *P. pennisetigena*; host, *C. ciliaris*; Morelos state; and not responsive to *MAT1* allele primers), CR1 (identified as probable *P. pennisetigena*; host, *C. ciliaris*; Tamaulipas state; and not responsive to *MAT1* allele primers) and DY1 (identified only as *Pyricularia* sp.; host, *D. ciliaris*; Morelos state; and not responsive to *MAT1* allele primers) were selected (Table 1). In general, all isolates were able to infect their original hosts (Table 3; Appendix A). However, the infection severity was different among the isolates of locations with different associated crops (Tamaulipas or Morelos). In *S. secundatum*, the SY2 and SR1 isolates (*P. oryzae*) displayed the classic eye-shaped morphology of leaf spots, whereas the CR1 isolate showed a few lesions without visible sporulation centers and a high relative severity scale of 4 (Figure 4). In *C. ciliaris*, the CY1 and CR1 isolates identified as probable *P. pennisetigena* produced eye-shaped blight lesions and sporulation centers, whereas the SR1 isolate identified as probable *P. pennisetigena* presented more reduced lesions without sporulation centers and a moderate severity (scale = 3). All *Pyricularia* isolates from *C. ciliaris* and *S. secundatum* grasses were able to infect *D. ciliaris* (Table 3). At the same time, the DY1 isolate, only identified as *Pyricularia* spp. from *D. ciliaris*, was not able to infect other host grasses. On *D. ciliaris*, the *P. pennisetigena* isolate DY1 presented oval-shaped blast lesions with necrotic centers, chlorotic borders and sporulation centers (Figure 4).

## 4. Discussion

Fifty-seven *Pyricularia* isolates were collected from grasses (*S. secundatum*, *C. ciliaris* and *D. ciliaris*) in different locations in Mexico. *Pyricularia* species have been reported to infect grasses in America [5,6,8,29]. In *S. secundatum* grass, several *Pyricularia* isolates, particularly *P. oryzae*, have been reported to be infecting agents [5]. In Mexico, there are also reports of this species infecting buffel grass (*C. ciliaris*) and rice [2,3,4]. These isolates have been generally identified by mainly using ITS regions [7,8]. In the current study, ITS sequence regions were obtained and analyzed in the *Pyricularia* isolates of this study. These sequences were added to an ITS dataset comprising *Pyricularia* accessions retrieved from GenBank to visualize and compare the sequences experimentally obtained. However, when a hotspot analysis was conducted, only one isolate from Brazil and the isolates of this study were displayed. This could be due to the lack of longitude/latitude data in the metadata of most of the GenBank accessions. We initially obtained diverse accessions from Argentina, Paraguay and the USA, but they were discarded in further analyses, as they did not contain indications of their regions of collection. However, the final ITS dataset covered 427 accessions (57 from this study) worldwide, a number representative of the ITS region in *P. oryzae* species. A network haplotype analysis of the ITS region showed a few redundant haplotypes; this suggests a tendency to homogeneity, which is a product of concerted evolution affecting the coding or non-coding regions of rDNA, often described in a wide range of taxa, including fungi [30]. However, when the network haplotype analysis was conducted only for the experimental sequences obtained in the current study, some non-redundant haplotypes related to *C. ciliaris* and *D. ciliaris* host grasses were visualized. These sequences also had a high identity with *P. pennisetigena* species, particularly for the isolates from the *C. ciliaris* grass. Usually, the ITS gene marker is firstly used, even in a multilocus analysis, to characterize *Pyricularia* isolates, since it often gives enough parsimony-informative characteristics to classify *Pyricularia* species [31]. However, in some cases, such as the one for *P. pennisetigena*, it is impossible to support a clear identification since it is considered a cryptic species within the *P. oryzae*/*grisea* species complex. This is mainly due to the fact that it is morphologically indistinguishable from *P. oryzae* [32,33,34], and this also cannot be resolved by using phylogenetic analyses [28]. Therefore, a more detailed analysis of the genetic variability among the isolates was conducted using a combined analysis of the polymorphic bands obtained from two genomic fingerprinting analyses and an ITS analysis in combination with microsatellites markers, which is an analysis that was proven to be useful in exploring the population diversity in *P. oryzae* from rice crops in Vietnam [14].

A Pot2-TIR analysis, which includes repetitive transposable element regions that are randomly distributed [17], was firstly conducted. The Pot2-TIR analysis has become one of the principal markers to study the genetic structure of field isolates of *Pyricularia* species worldwide, although it has only been applied to this fungal species [13,15]. In our case, the Pot2-TIR profile showed a high variability among isolates. The repetitive transposable element has been found to be involved in genomic variations among *P. oryzae* isolates [35]. Specifically, for the *P. oryzae* isolates from *S. secundatum*, this analysis formed a group comprising isolates from distant Mexican regions with different associated crops, and it had subclades with isolates specific to the state of Tamaulipas (associated with sorghum crops) or Morelos (associated with rice crops), suggesting a probable genetic adaptation to specific environmental regions, as the regions are distant (~1000 km) and exhibit different associated crops. The transposable Pot2-TIR element remains stable among different *Pyricularia* populations [36], and its copy number is modified by environmental changes or high geographical distances [35]. In general, the Pot2-TIR profile clearly separates isolates from different grasses and probably different *Pyricularia* species.

The *C. ciliaris* isolates, highly related to *P. pennisetigena* species, were clearly grouped by this analysis. The *Cenchrus* genus has been described as a host grass for this *Pyricularia* species [33]. To the best of our knowledge, there are no reports on the Pot2-TIR profiles of *P. pennisetigena* isolates. However, there are reports where the genetic flow between this species and the *P. oryzae* population has been demonstrated by using specific genes [28], a multilocus analysis [37] or comparative genomic analyses [38]. The authors of this last study observed a high proportion of LTR (12.6%) in the cryptic *P. pennisetigena* in relation to *P. oryzae* (<5%), which is relevant to the Pot2-TIR analysis, since transposable elements are located in LTR regions [23], and this probably explains why the application of this type of analysis might contribute to separating the cryptic *P. pennisetigena* from *P. oryzae.* Finally, the *Pyricularia* isolates from *D. ciliaris* could only be identified as *Pyricularia* sp. based on their ITS region, since they showed a clear difference with respect to other *Pyricularia* isolates.

Genomic fingerprinting was also conducted with ERIC-PCR profiles, which amplify randomly dispersed and non-repetitive regions on fungal genomes [18]. Therefore, a differential PCR profile aggrupation should be expected. In this study, ERIC-PCR showed less variability than Pot2-TIR, but it was useful to more clearly separate the isolates according to their host grass. This method also allowed us to separate the isolates according to their region of collection. Previously, this method has also been useful in separating *P. grisea* isolates in rice and weeds in Iran [39], and it has been used to assess the genetic diversity in fungal species [18,19]. Interestingly, ERIC-PCR was able to separate the *P. oryzae* isolates according to their *MAT1* allele. *P. oryzae* isolates share morphological characters, a low frequency of *MAT1-2* alleles and do not show reproductive structures. A low *MAT1-2* frequency and a lack of sexual recombination have also been observed for *Pyricularia* isolates from St. Augustine grass in different USA localities [40]. To the best of our knowledge, this is the first report of a relation between ERIC-PCR profiling and *MAT1* allele distribution in *P. oryzae* isolates, although more research is necessary to explore the occurrence of randomly amplified sequences and *MAT1* alleles. Taking both genomic fingerprinting analyses together, it is possible to support a high genomic diversity among the isolates of this study.

Diverse studies have related the genetic variability to the pathogenic profile in different *Pyricularia* isolates [15,16,17]. A spot infection analysis conducted to explore the pathogenic potential in these isolates was performed via agar plug inoculation, without punch damage to *S. secundatum* leaves. This analysis showed different infection profiles according to the proposed infection scale. GLS disease lesions have been well characterized in *S. secundatum* as brown spots with gray centers that can later expand across the entire leaf [41]. The differences were clearer when the damaged area was evaluated. Phenotypic assessments such as infection assays are the main types of analysis performed to assess the compatibility among host–pathogen and virulence [8,14]. However, artificial inoculation has been reported only in a few *Pyricularia* hosts [25,26]. Similar to this study, [42] evaluated some of these methods using 13 rice genotypes, and they found that the spot and filter paper inoculation methods were successful in discerning susceptibility to rice blast disease. In our study, the spot treatment with punction damage was, on average, 6.7 times more aggressive than that without punction, which might be related to the more effective induction of necrosis on the detached leaves analyzed, probably as a consequence of the mechanism of fungal infection in the plant tissues, which occurs through previous physical damage [43,44]. Additionally, we also evaluated the airbrush infection on detached leaves for representative *Pyricularia* isolates from *S. secundatum*. The results for the SY2 and SR1 isolates support their pathogenic profile, as previously observed with the spot infection assay. Airbrush infection was also used in a cross-infection assay with representative isolates from the *C. ciliaris* and *D. ciliaris* grasses obtained from the same locations. This analysis confirmed that all isolates could infect their original isolation hosts, but only a few isolates were able to infect other grasses. The isolates of *P. oryzae* from grasses can shift to other surrounding crops, such as rice and sorghum [9,10], probably due to their wide host range (Poaceae). Therefore, further research should be conducted to evaluate this possible cross-infectivity. The *Pyricularia* isolate (DY1) from *D. ciliaris* was not able to infect other grasses, and all isolates from *S. secundatum* and *C. ciliaris* were able to infect to *D. ciliaris.* Previous studies have shown that, in general, isolates from *S. secundatum* are unable to infect *D. sanguinalis* (crabgrass), and vice versa [6]. In general, isolates from *D. sanguinalis* are able to infect rice, green foxtail, common millet, barley and wheat, with variable percentages of success [1]. To the best of our knowledge, there are no reports on cross-infection between isolates from *S. secundatum* and *C. ciliaris.* These combined infection assays suggest a phenotypic variation in the studied isolates.

## 5. Conclusions

Fifty-seven *Pyricularia* isolates were isolated and characterized from infected commercial grasses *S. secundatum*, *C. ciliaris* and *D. ciliaris*, associated with sorghum and rice fields in two regions of Mexico. An ITS-based analysis (haplotype network) considering a dataset of ITS sequences of *Pyricularia* retrieved from GenBank allowed us to identify the isolates mainly as *P. oryzae*, which covers the redundant haplotypes, including the isolates from *S. secundatum*. Although a few isolates were obtained from other grasses, particularly from *C. ciliaris*, they were more closely related to *P. pennisetigena*. Genomic fingerprinting (Pot2-TIR and ERIC-PCR) showed a high variability among the isolates, which could be grouped by host grasses. Considering only the *P. oryzae* isolates from *S. secundatum*, these analyses allowed us to group them by region of collection, suggesting an adaptation process due to geographical distance. A phenotypic assessment carried out using a spot infection assay in representative *Pyricularia* isolates showed host switching between the isolates from *S. secundatum* and *C. ciliaris*.

## Figures and Tables

**Figure 1 jof-09-01055-f001:**
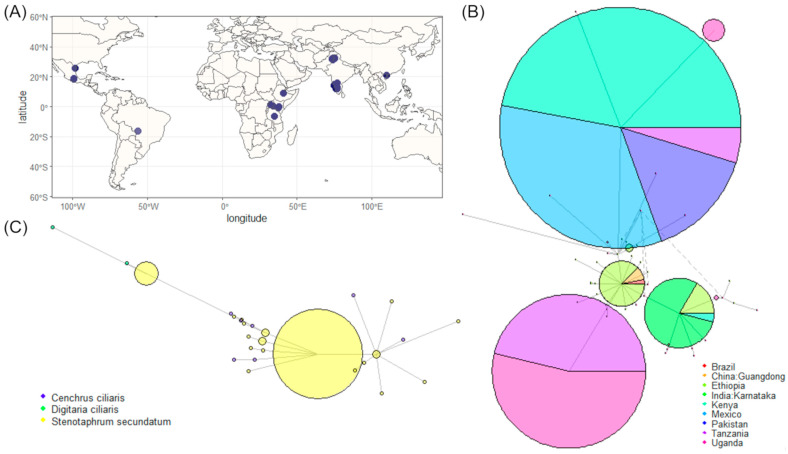
ITS region analysis of the *Pyricularia* accessions of this study and other accessions retrieved from GenBank: (**A**) hotspot localization, (**B**) haplotype ITS network. The red arrow shows the redundant haplotype that includes most of the *P. oryzae* of this study. Colors represent different geographical locations. (**C**) Haplotype ITS network built only with the *Pyricularia* isolates of this study. The red arrow shows the same redundant haplotype of panel B. Colors represent isolates of different host grasses. Lines indicate the different mutational steps. The circle area in the haplotypes is proportional to the ratio of each ITS sequence group.

**Figure 2 jof-09-01055-f002:**
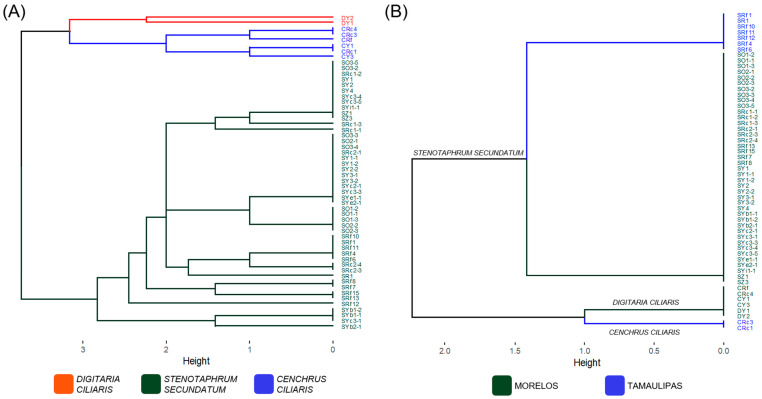
Genetic variability in the *Pyricularia* isolates in this study. (**A**) Pot2-TIR profile. UPGMA tree from absence/presence of band information. Color tree lines indicate the host origin of the isolates (red, *D. ciliaris*; green, *S. secundatum*; and blue, * C. ciliaris*). (**B**) ERIC-PCR profile. UPGMA tree from absence/presence of band information. Black tree lines are *P. oryzae* isolates. Green tree lines show isolates from the Tamaulipas region. Blue tree lines indicate the isolates from the Morelos region. Tree plots also indicate the specific clades for each host grass.

**Figure 3 jof-09-01055-f003:**
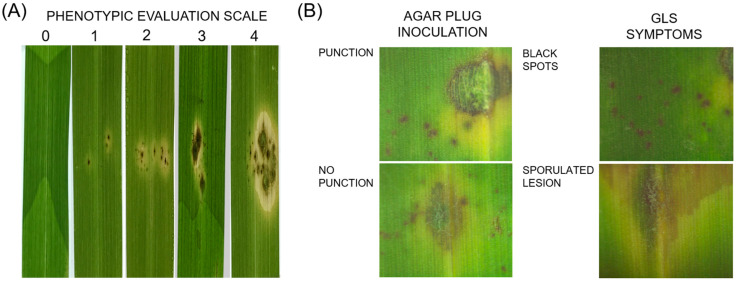
Infection symptoms produced by using representative *Pyricularia* isolates. (**A**) Phenotypic evaluation scale of GLS resistance on *S. secundatum*: 0 = no visible symptoms; 1 = dark brown, non-sporulating lesions with 1–2 mm diameter; 2 = expanding, dark brown, non-sporulating lesions; 3 = small circular or diamond-shaped lesions with sporulating areas; 4 = large, expanding lesions with sporulating areas. (**B**) GLS symptoms on detached leaves of *S. secundatum*. Above, agar plug inoculation with (left) or without (right) punction damage. Below, black spots (**left**) formed by independent infective colonies and a sporulated lesion (**right**).

**Figure 4 jof-09-01055-f004:**
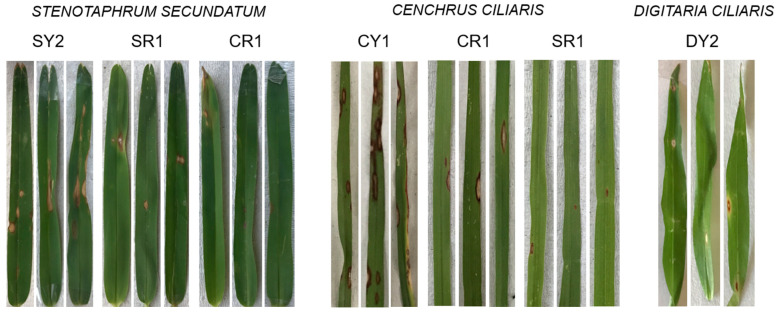
Cross-infection assay using representative *Pyricularia* isolates in three representative leaves of each grass. *P. oryzae* isolates from *S. secundatum* of the Morelos (SY2) and Tamaulipas (SR1) regions. *P. pennisetigena* isolates from *C. ciliaris* of the Morelos (CY1) and Tamaulipas (CR1) regions. *Pyricularia* sp. (DY1) isolate from *D. ciliaris* collected in the Morelos region.

**Table 1 jof-09-01055-t001:** *Pyricularia* isolates, identity, location and *MAT1* allele.

Accession Number	Isolate	Blast Identity (99%)	MAT1 Allele	Location	Host
MT785889	SY1	*P. oryzae*	Mat1-1	Yautepec, Morelos	*S. secundatum*
MT785890	SY2 ^a^	*P. oryzae*	Mat1-1	Yautepec, Morelos	*S. secundatum*
MT785891	SY4	*P. oryzae*	Mat1-1	Yautepec, Morelos	*S. secundatum*
MT785892	SZ1	*P. oryzae*	Mat1-1	Zacatepec, Morelos	*S. secundatum*
MT785893	SZ3	*P. oryzae*	Mat1-1	Zacatepec, Morelos	*S. secundatum*
OK185290	SO1-1	*P. oryzae*	Mat1-1	Oacalco, Morelos	*S. secundatum*
OK185291	SO1-2	*P. oryzae*	Mat1-1	Oacalco, Morelos	*S. secundatum*
OK185292	SO1-3	*P. oryzae*	Mat1-1	Oacalco, Morelos	*S. secundatum*
OK185293	SO2-1	*P. oryzae*	Mat1-1	Oacalco, Morelos	*S. secundatum*
OK185294	SO2-2	*P. oryzae*	Mat1-1	Oacalco, Morelos	*S. secundatum*
OK185295	SO2-3	*P. oryzae*	Mat1-1	Oacalco, Morelos	*S. secundatum*
OK185296	SO3-2	*P. oryzae*	Mat1-1	Oacalco, Morelos	*S. secundatum*
OK185297	SO3-3	*P. oryzae*	Mat1-1	Oacalco, Morelos	*S. secundatum*
OK185298	SO3-4	*P. oryzae*	Mat1-1	Oacalco, Morelos	*S. secundatum*
OK185299	SO3-5	*P. oryzae*	Mat1-1	Oacalco, Morelos	*S. secundatum*
OK185300	SRc1-1	*P. oryzae*	Mat1-1	Reynosa, Tamaulipas	*S. secundatum*
OK185301	SRc1-2	*P. oryzae*	Mat1-1	Reynosa, Tamaulipas	*S. secundatum*
OK185302	SRc1-3	*P. oryzae*	Mat1-1	Reynosa, Tamaulipas	*S. secundatum*
OK185303	SRc2-1	*P. oryzae*	Mat1-1	Reynosa, Tamaulipas	*S. secundatum*
OK185304	SRc2-3	*P. oryzae*	Mat1-1	Reynosa, Tamaulipas	*S. secundatum*
OK185305	SRc2-4	*P. oryzae*	Mat1-1	Reynosa, Tamaulipas	*S. secundatum*
MT785894	SR1 ^a^	*P. oryzae*	Mat1-2	Reynosa, Tamaulipas	*S. secundatum*
OK185306	SRf1	*P. oryzae*	Mat1-2	Reynosa, Tamaulipas	*S. secundatum*
OK185307	SRf4	*P. oryzae*	Mat1-2	Reynosa, Tamaulipas	*S. secundatum*
OK185308	SRf6	*P. oryzae*	Mat1-2	Reynosa, Tamaulipas	*S. secundatum*
OK185309	SRf10	*P. oryzae*	Mat1-2	Reynosa, Tamaulipas	*S. secundatum*
OK185310	SRf11	*P. oryzae*	Mat1-2	Reynosa, Tamaulipas	*S. secundatum*
OK185311	SRf12	*P. oryzae*	Mat1-2	Reynosa, Tamaulipas	*S. secundatum*
OK185312	SRf8	*P. oryzae*	Mat1-1	Reynosa, Tamaulipas	*S. secundatum*
OK185313	SRf7	*P. oryzae*	Mat1-1	Reynosa, Tamaulipas	*S. secundatum*
OK185314	SRf13	*P. oryzae*	Mat1-1	Reynosa, Tamaulipas	*S. secundatum*
OK185315	SRf15	*P. oryzae*	Mat1-1	Reynosa, Tamaulipas	*S. secundatum*
OK185316	SYb1-1	*P. oryzae*	Mat1-1	Yautepec, Morelos	*S. secundatum*
OK185317	SYb1-2	*P. oryzae*	Mat1-1	Yautepec, Morelos	*S. secundatum*
OK185318	SYb2-1	*P. oryzae*	Mat1-1	Yautepec, Morelos	*S. secundatum*
OK185319	SYc2-1	*P. oryzae*	Mat1-1	Yautepec, Morelos	*S. secundatum*
OK185320	SYc3-1	*P. oryzae*	Mat1-1	Yautepec, Morelos	*S. secundatum*
OK185321	SYc3-3	*P. oryzae*	Mat1-1	Yautepec, Morelos	*S. secundatum*
OK185322	SYc3-4	*P. oryzae*	Mat1-1	Yautepec, Morelos	*S. secundatum*
OK185323	SYc3-5	*P. oryzae*	Mat1-1	Yautepec, Morelos	*S. secundatum*
OK185324	SYe1-1	*P. oryzae*	Mat1-1	Yautepec, Morelos	*S. secundatum*
OK185325	SYe2-1	*P. oryzae*	Mat1-1	Yautepec, Morelos	*S. secundatum*
OK185326	SY1-1	*P. oryzae*	Mat1-1	Yautepec, Morelos	*S. secundatum*
OK185327	SY1-2	*P. oryzae*	Mat1-1	Yautepec, Morelos	*S. secundatum*
OK185328	SY2-2	*P. oryzae*	Mat1-1	Yautepec, Morelos	*S. secundatum*
OK185329	SY3-1	*P. oryzae*	Mat1-1	Yautepec, Morelos	*S. secundatum*
OK185330	SY3-2	*P. oryzae*	Mat1-1	Yautepec, Morelos	*S. secundatum*
OK185331	SYi1-1	*P. oryzae*	Mat1-1	Yautepec, Morelos	*S. secundatum*
OK185332	CR1 ^a^	*P. pennisetigena*	N.A.	Reynosa, Tamaulipas	*C. ciliaris*
OK185333	CRc1	*P. pennisetigena*	N.A.	Reynosa, Tamaulipas	*C. ciliaris*
OK185334	CRc3	*P. pennisetigena*	N.A.	Reynosa, Tamaulipas	*C. ciliaris*
OK185335	CRc4	*P. pennisetigena*	N.A.	Reynosa, Tamaulipas	*C. ciliaris*
OK185336	CRf1	*P. pennisetigena*	N.A.	Reynosa, Tamaulipas	*C. ciliaris*
OK185337	CY1 ^a^	*P. pennisetigena*	N.A.	Yautepec, Morelos	*C. ciliaris*
OK185338	CY3	*P. pennisetigena*	N.A.	Yautepec, Morelos	*C. ciliaris*
OK185339	DY1 ^a^	*Pyricularia* sp.	N.A.	Yautepec, Morelos	*D. ciliaris*
OK185340	DY2	*Pyricularia* sp.	N.A.	Yautepec, Morelos	*D. ciliaris*

^a^ *Pyricularia* isolates selected for further analysis. N.A. No amplification with the *MAT1* allele primers.

**Table 2 jof-09-01055-t002:** Establishment of the pathogenicity test in *P. oryzae* strains isolated from *S. secundatum* via agar plug inoculation with or without punction damage to detached leaves of *S. secundatum* grass.

Isolate	Location	Reaction Type Scale		Damaged Area (mm^2^)	
		Plug	Plug + Punction	Plug	Plug + Punction
SY1	Yautepec, Morelos	3	4	23.0 ± 3.6 ^b^	75.1 ± 9.5 ^a^
SY2	Yautepec, Morelos	4	4	76.7 ± 6.2 ^a^	73.6 ± 13.0 ^a^
SY4	Yautepec, Morelos	2	4	5.1 ± 2.3 ^c^	73.7 ± 6.2 ^a^
SZ1	Zacatepec, Morelos	1	4	1.8 ± 1.8 ^c^	15.4 ±7.0 ^c^
SZ3	Zacatepec, Morelos	1	4	1.6 ± 1.1 ^c^	17.6 ± 8.3 ^c^
SR1	Reynosa, Tamaulipas	3	4	24.07 ± 4.8 ^b^	42.1 ± 5.5 ^b^

^a^, ^b^ and ^c^ are different groups of according to the Tukey test (α = 0.05).

**Table 3 jof-09-01055-t003:** Cross-infection assay for *Pyricularia* isolates using an airbrush method on different host grasses.

Host	Representative Isolate	Leaf Lesions (Number) ^a^	Damaged Area (mm^2^)	Damaged Area (%)	SES Scale ^b^
*Stenotaphrum secundatum*	SY2	4.35 ± 1.1	57.57 ± 12.61	12.64 ± 2.58	6
	SR1	2 ± 0.79	9.82 ± 3.11	2.44 ± 0.87	4
	CY1	–	–	–	–
	CR1	2.1 ± 0.91	13.05 ± 5.04	2.92 ± 0.98	4
	DY1	–	–	–	–
*Cenchrus ciliaris*	SY2	–	–	–	–
	SR1	1.3 ± 0.57	3.52 ± 1.58	0.43 ± 0.31	3
	CY1	5.85 ± 2.21	80.45 ± 28.26	15.30 ± 3.21	6
	CR1	2 ± 0.65	16.41 ± 4.75	2.79 ± 0.78	4
	DY1	–	–	–	0
*Digitaria ciliaris*	SY2	–	–	–	–
	SR1	–	–	–	–
	CY1	–	–	–	–
	CR1	–	–	–	–
	DY1	2.4 ± 0.9	20.59 ± 3.56	3.17 ± 0.57	4

–: No infection. ^a^ At 7 days of infection. ^b^ According to [24].

## Data Availability

All data are available in the manuscript and in the Appendix A.

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
