# Peer review of "Pyricularia*’s Capability of Infecting Different Grasses in Two Regions of Mexico"

_jof, 2023, doi:10.3390/jof9111055_

Round 1
Reviewer 1 Report
1. At first glance, the manuscript has not been well checked before submission because there are still many obvious typos, and grammatical mistakes. The text and tables should be carefully revised. The scientific names should be italic
2. The study and its manuscript have been well conduced and structured. However, the abstract is difficult to follow. Using common names and brief explanations is a solution for this. For example, “Pyricularia genus are pathogenic fungi that infect different species of Poaceae, such as rice and sorghum” would be a more readable version. The same should be applied to not only the rest of the abstract but also other parts of the manuscript.
3. At the end of the abstract, a further application or contribution of the current study should be suggested.
4. Why are the three grasses (S. secundatum, C. ciliaris and D. ciliaris) chosen? There should have been a clearer explanation for the selection (the important of these grass in Mexico) and a more detailed introduction of these three. The potential risk of Pyricularia on other plant which is cultivated in surrounding area (Sorghum and rice)
5. I suggest dividing long paragraphs in the materials and methods into smaller ones.
6. Section 3.4, levels of infection should be mention in 2.5
7. More figures about the infection symptoms of the fungi on the plants and the growth and morphology of the fungi on petri dishes should be provided.
8. Make a conclusion in short
Minor editing of English language required
Author Response
Reviewer 1
Comments and Suggestions for Authors:
- At first glance, the manuscript has not been well checked before submission because there are still many obvious typos, and grammatical mistakes. The text and tables should be carefully revised. The scientific names should be italic.
Answer: The MS was grammatically revised to avoid mistakes. In specific, a detailed revision of the italic style in the scientific names was also done. Some long sentences were also shortened to improve the overall style.
- The study and its manuscript have been well conduced and structured. However, the abstract is difficult to follow. Using common names and brief explanations is a solution for this. For example, “Pyricularia genus are pathogenic fungi that infect different species of Poaceae, such as rice and sorghum” would be a more readable version. The same should be applied to not only the rest of the abstract but also other parts of the manuscript.
Answer: The authors agree with the reviewer. Redaction was carefully revised in order to improve the readability of the MS.
- At the end of the abstract, a further application or contribution of the current study should be suggested.
Answer: A sentence describing the main contribution of the current study was added in the abstract end (line 26).
- Why are the three grasses ( secundatum, C. ciliaris and D. ciliaris) chosen? There should have been a clearer explanation for the selection (the important of these grass in Mexico) and a more detailed introduction of these three.
Answer: A sentence was added in the methodology describing why these three grasses were chosen (Lines 198-199).
4.1 The potential risk of Pyricularia on other plant which is cultivated in surrounding area (Sorghum and rice).
Answer: The potential risk of Pyricularia isolates in grasses surrounding crop was firstly mentioned in the introduction (lines 40-42). We also added a brief discussion of this risk in the discussion section (Lines 560-562).
- I suggest dividing long paragraphs in the materials and methods into smaller ones.
Answer: The authors agree with the reviewer. Long sentences were shortened to improve the overall style, from line 62 to 111.
- Section 3.4, levels of infection should be mention in 2.5.
Answer: Methodology section was updated. Scala infection was mentioned in the section 2.5 although a detailed description of this was given in the result section.
- More figures about the infection symptoms of the fungi on the plants and the growth and morphology of the fungi on petri dishes should be provided.
Answer: A new supplementary Figure S3 showing the cross-infection in the different grasses grown in pots was given. In addition, a new Figure S1 shows the representative P. oryzae morphology of the isolates in petri dishes, details of the microscopic documentation were also given. Old Figure S1 was consequently renamed as Figure S2.
- Make a conclusion in short
Answer: Conclusion section was shortened. The authors thank the reviewer for the comments and suggestions given.
Comments on the Quality of English Language
- Minor editing of English language required.
Answer: The redaction of the MS was again revised to avoid mistakes and to improve the overall style.
Reviewer 2 Report
The article in general is well presented and interesting results are shown regarding the haplotypes, variability and infection capacity of Pyricularia.
My main observation is regarding the Haplotype Network analysis for which very few accessions have been used compared to other studies that use the same tool.
Regarding the references, there are some very generic and not relevant.
The writing and grammar in general are adequate. There are some errors that can be corrected after a review of the form.
Author Response
Reviewer 2
Comments and Suggestions for Authors
- The article in general is well presented and interesting results are shown regarding the haplotypes, variability and infection capacity of Pyricularia. My main observation is regarding the Haplotype Network analysis for which very few accessions have been used compared to other studies that use the same tool.
Answer: Two haplotype maps are presented, the first (Fig. 1B) includes all the sequences retrieved from Genbank with the geolocation criteria plus the study sequences (427 accessions); the second is specific to the Mexican accessions (Fig. 1C) obtained experimentally with 57 accessions. Although the number of 427 accessions in the first map (Fig. 1B) is relatively low, it is sufficient to resolve the redundant haplotypes, which are confirmed by the analysis carried out only with the Mexican accessions (Fig. 1C). The geolocation criterion has the advantage of also allowing accessions to be organized by country. Finally, we also included a maximum threshold for the number of mutations per haplotype linkage to simplify the descriptive drawing of the map, hence few lines of linkage between haplotypes are observed, even though there are 427 accessions in the global map (Fig. 1B). We include this methodological precision in the corresponding section (Lines 144-145).
- Regarding the references, there are some very generic and not relevant.
Answer: Four references that were redundant were eliminated, consequently the numbers in the references were updated.
Comments on the Quality of English Language
- The writing and grammar in general are adequate. There are some errors that can be corrected after a review of the form.
Answer: The MS was carefully revised to avoid grammatical errors.
Reviewer 3 Report
Intersting work on diveristy of Pyricularia species. Generally I have no objections to the quality of the description of methodology and results. Discussion is sufficient. Only small corrections are necessary.

Author Response
Reviewer 3
Comments and Suggestions for Authors
Interesting work on diversity of Pyricularia species. Generally, I have no objections to the quality of the description of methodology and results. Discussion is sufficient. Only small corrections are necessary.
Answer: The authors thank the reviewer for his/her comments and the MS was revised according to the suggestions to improve its quality and achieve the high standards of the journal.